# The Effect of Boarding on the Mental Health of Primary School Students in Western Rural China

**DOI:** 10.3390/ijerph17218200

**Published:** 2020-11-06

**Authors:** Bin Tang, Yue Wang, Yujuan Gao, Shijin Wu, Haoyang Li, Yang Chen, Yaojiang Shi

**Affiliations:** 1Center for Experimental Economics in Education (CEEE), Shaanxi Normal University, Xi’an 710119, China; tangbin@snnu.edu.cn (B.T.); wangyueceee@163.com (Y.W.); yujuan.gao@ufl.edu (Y.G.); wushijinceee@163.com (S.W.); haoyangliceee@163.com (H.L.); 2School of Economics and Finance, Xi’an Jiaotong University, Xi’an 710061, China; 3Food and Resource Economics Department, University of Florida, Gainesville, FL 32611, USA; 4School of Foreign Studies, Xi’an Jiaotong University, Xi’an 710061, China; a774130020@stu.xjtu.edu.cn

**Keywords:** boarding school student, mental health, rural China, difference-in-differences, matching

## Abstract

Based on the panel data of 20,594 fourth- and fifth-grade students in the western provinces A and B in China, this paper analyzed the effect of boarding at school on the mental health of students using a combination of the propensity score matching (PSM) and difference-in-differences (DID) methods. The results showed that boarding had no significant effect on the mental health of students, but the tendency of loneliness among boarding school students was increased. Heterogeneity analysis found that fifth-grade students whose parents had both left home to work were more likely to have poorer mental health when boarding. This paper has essential policy significance for guiding rural primary schools to improve the mental health of boarding school students, especially left-behind children.

## 1. Introduction

For individuals, receiving good education is conducive to the improvement of individual human capital and income level; For the country, the development of education is also conducive to the overall level of human capital and economic transformation and development [1,2]. Many developing countries in the world have attached importance to the construction of school institutions at the basic education stage. For example, Duflo (2001) conducted an empirical study on the large-scale establishment of schools in Indonesia in the last century showing that if there is one more primary school for every 1000 children on average, the average years of schooling will increase by 0.12–0.19 years [3]. Berlinski, Galiani, and Gertler (2009) based on Argentina’s study also found that the construction of pre-school education institutions significantly contributed to primary school students’ academic performance [4].

China’s rapid progress in education in recent decades has benefited from its long-term investment in basic education. In the 1990s, China implemented the national compulsory education project and realized the pattern of “one village, one school” nationwide. However, since the mid-1990s, the number of school-age children in rural areas continued to decline, “one village, one school” school scale becoming small, and high management costs, staffing difficulties, and other issues were increasingly prominent [5]. The change of population situation brought about increase of management cost and difficulty of rural primary schools, which became the motivation of rural school layout adjustment. In 2001, China’s government issued the “Decision of the State Council on the Reform and Development of Basic Education” and proposed “adjusting the layout of rural compulsory education schools in line with local conditions”, which started a new round of adjustment of the layout of primary and secondary schools in rural areas, also known as the “withdrawal and consolidation of schools” [6].

In 2001, the State Council’s Decision on Reform and Development of Basic Education pointed out that the layout of rural compulsory education schools should be adjusted according to local conditions [7]. Since then, the movement “withdrawal and consolidation of schools” began and swept across China’s rural primary and secondary schools. The implementation of this policy to some extent did integrate rural education resources, reduce the cost of education per student, but also increased the number of boarding schools [8]. Since 2001, with the continuous development and in-depth reform of China’s education, as well as the needs of a large number of left-behind children in rural areas whose parents leave to work in urban areas, boarding schools have developed rapidly in rural areas of China and thus the number of boarding school students has quickly increased [9]. By the end of 2015, the total number of boarding school students across rural primary and secondary schools reached 26.363 million. The boarding rates of primary school and junior high school students were 14.4% and 58.6%, respectively. In the western rural areas, the proportion of primary school boarders reached 21.1% [10].

Although it is relatively easy to monitor the physical health status of students through more frequent and regular school physical examinations, it is more difficult to detect levels and problems of mental health [11]. A series of devastating suicides cases due to mental health problems from students in the early 2000s have prompted the Chinese government to make considerable efforts to reduce mental health problems in schools at all levels [12,13]. China’s government attaches great importance to the development of mental health among adolescents. The “Guidance on Strengthening Mental Health Services” report issued in December 2016 pointed out that “in order to comprehensively strengthen mental health education for children and adolescents, primary and secondary schools should pay attention to students’ mental health education, cultivate positive, optimistic, and healthy psychological qualities, and promote the sustainable development of students’ mind and body” [14]. The rapid development of the boarding system in rural primary and secondary schools in China has increased concern and discussion regarding boarding school students’ mental health.

Theoretically, boarding has both beneficial and detrimental effects on adolescent growth. On the positive side, boarding, a kind of collective life, to some extent, may reduce the risk of psychological problems on students. For example, group living environment supervised by school-assigned student caregivers may help maintain the health and safety of boarding students [15]. For those students who do not have enough family care, they may be able to get better care and life at school than in their families, which may contribute to their healthy development [16]. Moreover, psychological counseling from teachers and communication with peers can help disadvantaged students to overcome the troubles and psychological problems and other common challenges experienced by students [17]. On the negative side, boarding implying the lack of care from family may worsen students’ mental health. Students in the primary school education stage are in a critical period of growth and development and boarding school students are separated from their parents for a long time, thus it is difficult for them to receive daily care from their parents and families during that period [18]. The collective living environment of boarding schools is also likely to resulting in students being more affected by the bad habits and behaviors of other students, and even makes them more susceptible to suffering from bullying, both of which would have a long-term negative impact on students’ physical and mental development [19,20]. Serious mental health problems may hurt students’ academic performance, which is not conducive to the improvement of China’s human capital [21].

In recent years, the number of boarding schools in China has increased rapidly and a few studies have paid attention to the effect of boarding on students’ mental health status. However, from empirical literature, research on the impact of boarding on students’ mental health is inconsistent. A couple of studies have shown that the impact of boarding on school students’ mental health is negative. Evans-Campbell et al. (2012) believed that individuals that have been enrolled in boarding schools or have been raised by a person attending a boarding school are more likely to develop significant anxiety disorders and post-traumatic stress disorder than other individuals, and are even more likely to develop suicidal ideations [22]. Another study has shown that boarding hurts students’ mental health, and that the negative effects do not disappear due to short-term [23]. Ma et al., (2013) found that the overall mental health of boarding school students was poor by analyzing the variances in fifth- and sixth-grade primary school students. Primary school students’ self-blame tendency, learning adaptation independence, learning anxiety, allergy tendency, physical symptoms, and mental health test (MHT) total scores were significantly higher than those of non-boarding primary school students [24]. Chen et al., (2020) adopted instrumental variable to examine the effect of boarding and found to have negative effects on a few dimensions of students’ mental health (i.e., study anxiety, social anxiety, self-punishment, physical anxiety symptoms, and fear) reaching at 0.455 SDs (standard deviations) [11]. In contrast, some studies have shown that boarding either has no significant effect or has uncertain effects on the mental health of students. For example, Shen et al., (2012) used first- and second-year students in junior high school as research samples. He used multiple regression analysis to determine that there was no significant difference in the average scores of the full-scale MHT(Mental Health Test) and the eight subscales between boarding and non-boarding school students indicating that there was no significant difference in the mental health status between boarding and non-boarding school students [25]. Liu and Villa (2020) detected that though students who board in schools have improved their academic scores, their mental health level changed little [26]. Liu et al. (2004) used the psychotic symptom self-assessment form (SLE-90) to evaluate boarding and non-boarding high school students. They found that although boarding school students have more psychological problems in the initial enrollment stage, the psychological problems in the upper grades gradually decrease and the psychological health increases [27]. 

Overall, the research literature in this area is quite rich, but there are still a few obvious problems that need to be further solved: First, the research methods used are mostly one-dimensional or multiple regression models and cross-sectional data are mostly used, which fail to solve the endogeneity problem caused by the self-selection bias and the omitting relevant variable bias. For example, Ma et al. (2013) and other researchers studied only 900 fifth- and sixth-grade students and adopted a simple variance analysis method. Second, the definition of boarding in most of the literature only examines whether students are boarding at a certain time and does not consider the change in boarding status of students across a period of time. For example, Shen et al. (2012) used multiple regression and distinguished between boarding and non-boarding school students purely based on each student’s boarding status during the survey. Since the time at which different rural children can board is different, it is reasonable to suspect that those students who board earlier are likely to experience other exogenous events in later school life. Without identification and divestiture, it is likely that the negative effects of boarding behavior will be overestimated [27]. Another important deficiency is that the existing research samples are mostly from either a single or a few schools in a single province, and the sample sizes are small and non-representative. This paper will try to cope with the deficiency list above using large, representative data and a combination of Propensity Score Matching (PSM) and the Difference-in-Differences (DID) model to overcome endogenous problems.

Based on the panel data of 20,594 fourth- and fifth-grade students in the western provinces A and B, this paper used a combination of Propensity Score Matching (PSM) and the Difference-in-Differences (DID) model to correct the endogenous problems that may exist in the model, and to provide evidence for the effect of boarding on the mental health of rural primary school students.

The rest of the paper is organized as follows: Section 2 introduces the data and empirical models used in this paper; Section 3 shows the results of the model, including the impact of boarding on the mental health of primary school students, and heterogeneity analysis of the results; Section 4 provides the conclusion and policy recommendations.

## 2. Data and Methodology

### 2.1. Data

The tools and data used in this study were from a sample survey conducted by the Center for Experimental Economics in Education of Shaanxi Normal University in the western provinces A and B (We are using A and B instead of the provinces’ names for anonymity purposes). The survey used a stratified random sampling method. First, 11 counties in a certain city of province A were selected, 7 counties in a certain city of province B were selected, and then 252 rural primary schools were randomly selected from the sample counties. A total of 20,594 students in the fourth and fifth grades were randomly selected. The research team conducted a baseline survey and a follow-up survey of a sample of the 20,594 students in September 2012 and May 2013, respectively. Since some students had transferred to other schools or did not present on the day of the follow-up survey, part of the sample was lost, and finally, 16,685 fourth- and fifth-grade students from 251 schools participated in the follow-up survey. In order to make out the difference between retained samples and lost samples, we conducted a comparison check between two parts of sample. As shown in Table 1 the mean and SD of most students’ characteristics between retained samples and lost samples are similar though some characteristics show difference (gender, grade, family assets, baseline math score).

### 2.2. Data Description and Variables

A standardized questionnaire survey was conducted among students in the fourth and fifth grades. The questionnaire collected the following information from the students: (1) mental health, (2) boarding status, and (3) family background and socioeconomic status.

#### 2.2.1. Students’ Mental Health

This paper used the MHT to examine students’ mental health. This test was devised by Bucheng Zhou, a professor of the psychology department at East China Normal University, based on the “Uneasy Propensity Diagnostic Test” compiled by Suzuki Kiyoshi et al, from Japan [28]. The internal consistency coefficient of the full scale is 0.91, which has been widely used in the measurement of mental health of school-aged children in China.

The test has a total of 100 “yes” or “no” questions; selecting “yes” corresponds to 1 point, while “no” to 0 points. The 100 test questions can be divided into two parts, namely, the lying scale and the content scale. The lying scale consists of 10 questions; the higher the score, the higher the probability that the student will provide false information when completing the test. In this study, questionnaires with a score higher than 7 on the lying scale were invalidated (The samples used for analysis later have eliminated the samples which were invalidated). The content scale includes 8 sub-scales: learning anxiety, anxiety about people, the loneliness tendency, the self-blame tendency, the allergic tendency, physical symptoms, the horror tendency, and the impulsive tendency. The higher the sub-scale score, the worse the student’s performance in this dimension. The total score of 8 eight sub-scales indicates the overall level of mental health of the students—a high score corresponds to higher risk for mental health problems. This paper used standardized MHT scores to measure the students’ level of mental health.

The mean of the MHT standardized scores of the students in the sample is shown in Figure 1. It can be seen that in 2012 and 2013, levels of mental health between boarding students and non-boarding students are almost same. Figure 2 and Figure 3 show the distribution of the MHT standardized scores in the samples from 2012 and 2013.

#### 2.2.2. Students’ Boarding Status

This paper aimed to examine the impact of boarding on students’ mental health through the changes in the two-year data. Therefore, students who did not board in the baseline period were selected (13,638 samples). Students who were non-boarding in the baseline period (2012) would be in two states in the follow-up period (2013): boarding in 2013 (1020 samples) and non-boarding in 2013 (12,618 samples). This paper aimed to find out the impact of boarding through the difference between boarding students and non-boarding students in the second year. In this way, we can estimate the net effect of a year of boarding on students’ mental health, excluding other factors affecting mental health. For example, if we do not rule out the baseline period of boarding samples, it will be not clear that the change of mental health is stemmed from one-year boarding or from other aspects since students have been boarding before.

To ensure that all students at baseline had the same boarding status (non-boarding), this paper removed 3047 boarding school students from the baseline and finally collected 13,638 samples from the remaining 200 schools for research. According to the boarding status of the remaining sample in the follow-up period, in this paper, the students who were not boarding in the baseline period (2012) but were boarding in the follow-up period (2013) were defined as the treatment group. In addition, students who were not boarding in the baseline (2012) or follow-up (2013) periods were defined as the control group. As can be seen in Table 2, 81.74% of the students in the sample were not boarding in the baseline period, among which 75.62% were still not boarding during the follow-up period, and 6.11% of the students started boarding during the follow-up period.

#### 2.2.3. Students’ Family Background and Socio-Economic Status

In addition to boarding status, the student questionnaire collected basic personal and family information. The students’ basic personal information included gender, age, grade, whether or not they had myopia, their mathematics scores at baseline, and the distance from the school to the county. Before the questionnaire survey, the research team tested the students with internationally used mathematics test questions, and the investigators controlled the time and order on the spot to prevent cheating. The family information mainly included the family’s economic situation, the education level of parents, and the parents’ migration status for work. In the questionnaire, the students were asked whether their family had a business, whether they had computers, whether they had internet access, and whether they had electric cars, automobiles, running water, microwave ovens, refrigerators, cameras, video cameras, washing machines, and flushing toilets. Family assets were measured by the factor score, which was calculated by factor analysis according to the students’ answers to the above questions. 

There were some significant differences in individual and family characteristics between the treatment and control groups (Table 3). By directly comparing the individual and family characteristics of the treatment and control groups, it was found that, first, in terms of the individual characteristics of the students, the treatment group was mostly male and their mathematics scores at the baseline were lower. Second, as far as the family background is concerned, the students in the treatment group had a better family financial situation, the mothers were less educated, and the proportion of parents migrating to urban areas for work was low. In addition, the school where the treatment group was located was far from the county.

In summary, there were some significant differences between the treatment and the control groups at the individual and family levels. These differences may not only be related to students’ boarding status, but also to students’ mental health level. Therefore, it is necessary to control these factors and the endogenous problems generated by them when analyzing the impact of boarding on students’ mental health level.

## 3. Method

### 3.1. Propensity Score Matching (PSM)–Difference-in-Differences (DID) Method

According to Table 3, there were significant differences in individual characteristics and family economic background between the students in the treatment and control groups. These differences led to endogenous problems in the study, such as missing variable errors caused by unobservable factors and self-selection errors caused by observable factors. Therefore, this paper attempted to solve the above two problems by using a combination of propensity score matching (PSM) and the difference-in-differences (DID) model.

The DID model is a quantitative statistical method, which is applied to at least two periods of data. It simulates the samples with changed research behavior between two periods of data as the “experimental group” in the real experiment and the samples without change as the “control group” in the real experiment to explore the impact of the changes in research behavior on the outcome variables [29,30]. Specifically, in this study, when exploring the effect of boarding on students’ mental health, we chose students who did not board in the baseline period as the analysis sample (N = 13,638), and divided this sample into two groups: one group consisted of students who changed from non-boarding to boarding, as the “experimental group” (N = 1020); the other group consisted of the students who maintained their non-boarding status, as the “control group” (N = 12,618). The idea of applying DID method is to compare the difference in the change of an output variable between the experimental group and the control group before and after the implementation of a policy or before and after a certain behavior change.

The change in boarding behavior may be self-selection rather than random occurrence. If it is not treated and the sample is directly regressed, the parameter estimation will be biased. Economists proposed the propensity score matching method to reduce the error problem in the observation data set [31]. To avoid possible selection bias, propensity score matching was used in this paper. The basic idea of the PSM is to find one or several control group students with similar or even the same endowment characteristics for each experimental group student through PSM, and then to compare the mean values of the result variables of the experimental and control groups under by controlling the other variables to obtain the estimation of the impact of boarding on the students’ mental health [32]. The estimation steps involve: (1) the propensity scores of the students were estimated to establish the experimental group; (2) according to the common support of the propensity scores, the students in the experimental and control groups were matched by the closest matching method; and (3) a balance test was carried out on the two matched groups of samples.

In this paper, the DID method controlled the influence of unobservable variables through the difference in mental health before and after the change of boarding between the experimental group and the control group, especially the influence of factors that do not change over time and synchronously change over time, so as to effectively evaluate the net effect of boarding on students’ mental health. However, since the experimental group and the control group had different student and family characteristics, which does not meet the common trend premise assumption of using DID method, it was necessary to select the samples with similar characteristics in the control group and the experimental group as the counterfactual before performing DID. PSM can weaken selection bias and can obtain comparable samples of treatment and control groups. However, only observable factors can be controlled, and there are still missing variables for unobservable variables [33]. Therefore, this paper combined the PSM and DID methods to reduce endogeneity, since the DID method can control unobservable individual effects that are invariant over time. DID was used to estimate the average treatment effect of boarding on students’ mental health based on the PSM results, with the differences in the standardized MHT scores as the explained variable. Therefore, this paper used PSM–DID to estimate the average effect of boarding on students’ mental health.

### 3.2. Models

Based on the above principles, in order to analyze the impact of boarding on students’ mental health, the model in this paper was set as follows:(1)$$\Delta MHT_{is}=\alpha +\beta \cdot Boarding_{is}+\delta \cdot MHT_{is,baseline}+\gamma \cdot X_{is}+\varepsilon_{is}$$
where *i* indicates the student; *s* indicates the school; $\Delta MHT_{is}$ refers to the change in students’ MHT standardized scores in the follow-up period compared to the baseline period; $Boarding_{is}$ is the intervention variable. $Boarding_{is}$ = 1 indicates that the students are in the experimental group, that is, the students were not boarding in the baseline period but were boarding in the follow-up period; meanwhile, $Boarding_{is}\;$= 0 indicates that the students are in control group, that is, the students were not boarding in either the baseline or the follow-up period.

In this study, the important variables that may affect whether students board or not and the students’ mental health level were controlled. $X_{is}$ in the model indicates the control variables: (1) basic personal information: Gender, age, grade, whether the students are myopic, standardized mathematics scores of the students at baseline, and the distance from the school where the students are located to their home county; (2) family background and socio-economic status: family income, parents’ education level, and whether both parents are migrant workers. $MHT_{is,baseline}$ indicates the standardized mental health level of students at baseline. Through the value of the coefficient *β* and its significance, we were able to analyze the impact of boarding on students’ mental health.

In order to further analyze the impact of boarding on students’ mental health, we also analyzed the impact of boarding on eight dimensions of mental health (learning anxiety, anxiety about people, loneliness tendency, self-blame tendency, allergy tendency, physical symptoms, horror tendency, and impulse tendency). The specific model settings were as follows:(2)$$\Delta y_{is}=\alpha +\beta \cdot Boarding_{is}+\delta \cdot y_{is,baseline}+\gamma \cdot X_{is}+\varepsilon_{is}$$
where $\Delta y_{is}$ represents the scores of the standardized sub-scales for students in the follow-up period, and $y_{is,baseline}$ represents the scores of the standardized sub-scales for students at baseline.

Considering that for students with different characteristics, the impact of boarding on their mental health level may be different, we also analyzed the heterogeneity of the results from the perspective of students’ personal characteristics, family background, and socio-economic status. The specific model was set as follows:(3)$$\Delta MHT_{is}=\alpha +\beta \cdot Boarding_{is}\cdot H_{is}+\delta \cdot MHT_{is,baseline}+\gamma \cdot X_{is}+\varepsilon_{is}$$
where $H_{is}$ represents the basic characteristic variables of each individual (i.e., the gender, age, grade of the students, whether the students are myopic, and the distance from the school where the students are located to the county), as well as various family background and socioeconomic status variables (i.e., family income, parents’ education level, and whether both parents are migrant workers).

## 4. Results and Discussion

### 4.1. The Effect of Boarding on Students’ Mental Health

After matching based on the personal and family characteristics of the students, the changes in the MHT standardized scores of the students in the experimental and control groups were found to differ by a standard deviation of 0.02 (Table 4, Row 1, Column 1), but the difference was not significant. This shows that boarding has no significant effect on students’ mental health. According to the various dimensions, there was a significant difference between the students in the experimental group and the students in the control group only in the standardized scores of loneliness tendency, differing by a standard deviation of 0.32 (Table 4, Row 4, Column 1). This shows that after boarding, there is no significant change in the other mental health dimensions except that the loneliness tendency increased after boarding.

As for this result, there are three possible explanations.

First, in recent years (before or around 2012), China has attached great importance to the development of the mental health of primary and secondary school students and has increased its investment in boarding schools. For example, a few policies have been implemented to help students and schools to build up mental health education, such as “Guidelines for Mental Health Education in Primary and Secondary Schools” issued by Ministry of Education in 2002 and 2012. This policy helps teachers to help students with mental health disorders recover and adjust, especially for boarding students. Schools often play a substitute role for parents in terms of supervision and psychological counseling of students to some extent. However, due to students’ natural psychological attachment to their parents, schools cannot completely replace the role of parents. Studies such as that of Li et al., (2015) have shown that parent–child attachment has a direct impact on boarding school students’ adaptation to school [34]. Parent–child attachment refers to an intimate and lasting emotional connection with parents, which can provide support, security, and self-confidence for individuals. Boarding school students are far away from their parents and family for a long time and lack security, which makes it easy for them to feel lonely, which is a kind of explanation why loneliness tendency is significant higher. 

Second, left-behind children accounted for a high proportion. In 2012, there were about 165 million migrants and 44.76% of Chinese left-behind children (LBC) aged 12 to17 lived with their grandparents [35]. If we analyze the impact of boarding and left-behind children, as shown in Table 5 last row, the mental health of boarding LBC is worse than that of non-borders, which has an impact of 0.29 standard deviation and is significant at the confidence interval of 5%. The result implies if one parent goes out for working and students are boarding, their mental health is worse than who are not boarding, which also shows that if parents are not at home, students are prone to psychological problems such as loneliness. The above result is consistent with a latest study published in 2020. Chen et al. (2020), using data of 7606 rural students, proved that the effect of boarding is significantly higher among disadvantageous students. This study uses Instrumental Variable to examine the casual effect.

Third, in order to accurately identify the causal effect of boarding, this paper regards boarding as an intervention, which merely lasts one year. Short boarding period (one-year boarding) may not be enough for students’ mental health change.

Though the above conclusions are contrary to the research results of the existing literature, there might be some deficiency of the previous research. Specially, most research on the relationship between boarding and students’ mental health states that boarding has a negative effect on students’ mental health, but such research has some problems, such as small sample sizes, unrepresentative samples, and unresolved endogeneity. For example, Zhang et al. (2009) used the mental health scale for middle school students as a test tool to compare the mental health status of 274 junior high school students in boarding schools and 300 junior high school students in non-boarding schools in Ningxia, China [36]. The study found that the overall psychological problems of junior high school students in boarding schools were significantly higher than those of non-boarding schools, and the detection rate of various psychological problems in the former was also higher than that of the latter. However, the sample size of this study was too small, and the method adopted was a simple *t*-test, which failed to solve the problems of endogeneity. Moreover, the research object of this paper was junior high school students, and there may be some differences between their mental health and that of primary school students. Another study from Wang and Mao (2015) analyzed the survey data of 8047 primary school students in grades 4, 5, 7, and 8 in 11 western regions [37]. They believed that boarding did not play a substitute role for family supervision for left-behind children, but rather became a negative factor affecting the development of left-behind children’s social–emotional ability. Although the sample size was large and representative, the method used in this study was simple multiple regression analysis, and the mental health level of the students was tested through the self-compiled “Primary and Secondary School Students’ Social Emotional Ability Questionnaire”. Although the scale passed reliability and validity tests, the reliability of the scale needs further examination. This may be the deviation caused by different measuring tools. However, this paper has adopted a more convincing method PSM–DID with large and representative data, which can clearly estimate the real impact of boarding. We assume the results have been similar with most recent updated studies, such as Shi et al. (2016), Chen et al. (2020), and Liu and Villa (2020) [11,26,35]. 

### 4.2. Heterogeneity Analysis of the Effect of Boarding on Students’ Mental Health

There was no significant change in the mental health level of the students after boarding, but considering the different characteristics of the students, the changes in the mental health level after boarding may be different. This paper analyzed the heterogeneity of the impact of boarding on the mental health of the students, and the results are shown in Table 6. The results show that there is heterogeneity in the impact of boarding on students’ mental health in terms of grade and whether their parents are migrant workers. However, for the other individual and family characteristics, there is no heterogeneity in the impact of boarding. In order to further analyze the different effects of boarding on the mental health of different groups of students, this paper used model (1) to study the changes in the MHT standardized scores of the students after boarding. The results are shown in Table 7.

First of all, according to the results in rows 1 and 2 of Table 5, the students in the experimental group of grade 4 had a better mental health level than those in the control group, while the students in the experimental group of grade 5 had a significantly poorer mental health level than those in the control group. That is, the mental health level of fourth-grade students improves after boarding, while the mental health level of fifth-grade students deteriorates. To explore the possible explanation, we made regression of the boarding effect on sub-dimensions of mental health between grade 4 and grade 5 as shown in Table 7. It can be seen that among students who have been boarding, in the dimension of learning anxiety and anxiety about people, the anxiety level of the fifth grade students is obviously higher than that of the fourth grade students, amounting to 0.39 SD and 0.27 SD. This may be because as they reach higher classes, the students find it more difficult to learn the course; and the fifth-grade students are facing greater pressure than the fourth-grade students. Studies have proven that in rural schools students do not have a solid learning foundation. With the increase in grade, the learning content becomes more and more complex, the knowledge that students are not able to fully master increases, and their learning anxiety worsens [38]. Additionally, fifth-grade students are facing great pressure to enter junior high school, and their mental health level is poorer [35]. Therefore, the mental health level of fifth-grade students deteriorates significantly after boarding.

Second, according to the results in rows 5 and 4 of Table 5, for families where at least one parent has not left to work in an urban area, there was no significant change in the mental health level of the students after boarding. However, for families whose parents are both migrant workers, the mental health level of students after boarding was significantly worse (SD = 0.30) at a significance level of 5%. If one parent is at home, boarders can avoid psychological problems by communicating with their parents when they return home on weekends. However, for families where both parents migrate to urban areas for work, students are not well cared for by their parents. After boarding, students are more likely to have psychological problems due to their reduced contact with their families, which is in line with literature in left-behind children, such as, Ge et al. (2015), Fellmeth et al. (2018), Jia and Tian (2010), and Zhao and Yu (2016) [39,40,41,42]. There is also possibility that families where both parents need to go out have unobserved differences to other families (i.e., financial issues, opportunity structure). Overall, it might be that boarding can make up for the lack of family supervision of left-behind children to some extent, but it cannot completely replace the emotional communication of families.

## 5. Conclusions

Based on the panel data from 2012 to 2013, this paper used the DID and PSM methods to estimate the impact of boarding on the mental health of rural primary school students. It was found that boarding had no significant effect on the mental health of rural primary school students, but the tendency of loneliness after boarding increased. There were significant differences in the mental health of students between grades. There was also a significant difference in the mental health of students depending on whether or not their parents migrated for work.

Unlike most existing studies, this study concluded that there was no significant difference in the mental health of boarding school students. There are three possible reasons. One possible reason is that in recent years, China has attached great importance to the development of adolescents’ mental health and has increased its investment in the country’s boarding schools. An example of the importance attached to boarding schools can be found in the “Guidelines of the State Council on Strengthening the Care and Protection of Left-behind Children in Rural Areas” released in 2016, which proposed that the administrative department of education should support and guide small- and medium-sized schools to strengthen mental health education, promote the positive and healthy development of adolescents’ mental health and personalities, and both detect and correct any psychological problems and bad behaviors early. The document also proposed to “strengthen the construction of rural boarding schools, promote the rational distribution of boarding schools, and meet the schooling needs of left-behind children in rural areas.” This work was listed as the main point of work for the Ministry of Education for 2016 and 2017 [43]. Therefore, schools could be developed to replace the supervisory role of the family to a certain extent and can provide a certain degree of psychological diversion. However, due to students’ natural attachment to their parents and the lack of family communication for boarding school students, school cannot completely replace the role of parents. Therefore, in addition to focusing on cultivating the independent living ability of boarding school students, schools should also pay attention to mental health issues such as loneliness as a result of boarding. The second reason that cannot be ignored is that left-behind children still account for a high proportion, so it is reasonable that loneliness tendency has been significantly higher among eight dimensions of MHT. In addition, it is also possible that the impact of one year’s boarding on students has not yet appeared. It is recommended to shorten the holiday period as appropriate to meet the students’ desire to get warmth and emotional connection from family ties. At the same time, schools should pay attention to the development of students’ mental health, implement relevant courses, and hire professional mental health teachers to provide psychological counseling to students.

Compared with the existing research, the advantage of this paper is that it received follow-up survey data from large and representative samples of rural students. Furthermore, the panel data were used to solve the endogeneity problem caused by the self-selection bias and missing variables of the sample through the methods of PSM–DID. However, there are still drawbacks in this paper: (1) the problem of self-selection bias caused by unobservable factors could not be completely solved by PSM; (2) it was difficult to solve the mutual causation problem between boarding decisions and students’ mental health levels. However, this paper still supplements the existing literature and provides a new dimension of evidence for understanding the impact of boarding on students.

## Figures and Tables

**Figure 1 ijerph-17-08200-f001:**
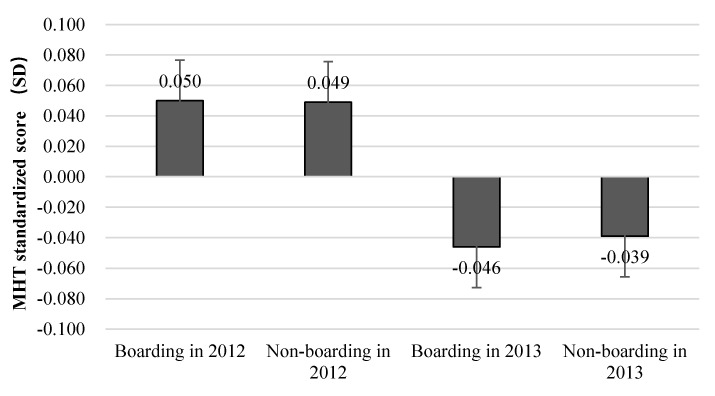
Mean of standardized students’ mental health test (MHT) scores.

**Figure 2 ijerph-17-08200-f002:**
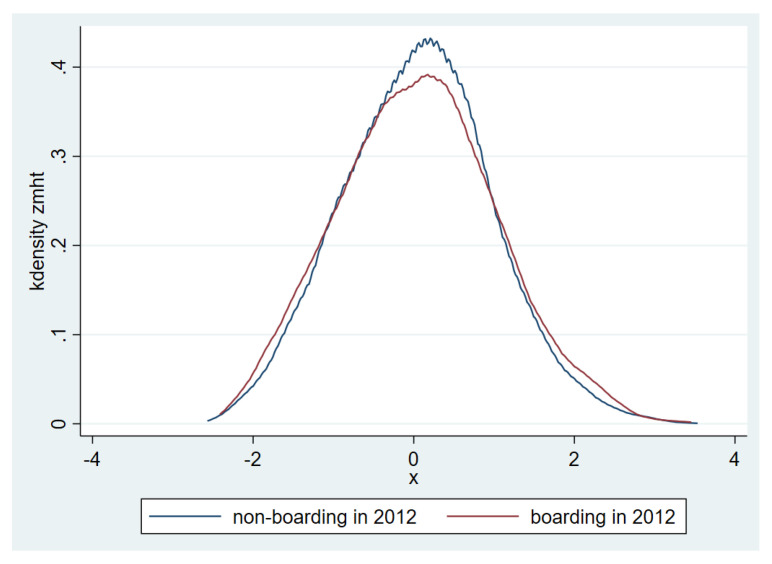
Distribution of students’ standardized MHT scores in samples from 2012.

**Figure 3 ijerph-17-08200-f003:**
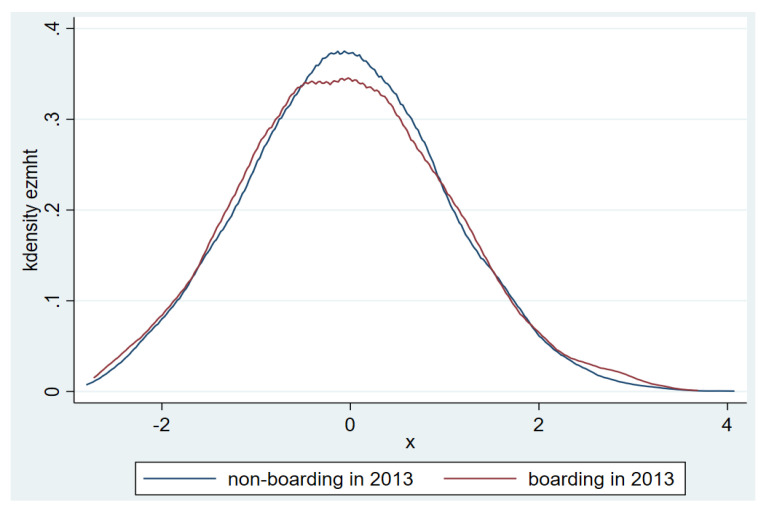
Distribution of students’ standardized MHT scores in samples from 2013.

**Table 1 ijerph-17-08200-t001:** Comparison of the characteristics of students between retained samples and lost samples.

Control Variables	Retained Samples	Lost Samples	H0: (1) = (2) Difference
	Mean	Mean	Mean
	(SD)	(SD)	(*p*-Value)
	(1)	(2)	(3)
Students’ characteristics			
(1) Age (1 = at least 10 years old; 0 = less than 10 years old)	0.83	0.87	0.04
	(0.37)	(0.34)	(0.35)
(2) Gender (1 = male; 0 = female)	0.51	0.56	0.05 ***
	(0.50)	(0.50)	(0.00)
(3) If the student is in grade-4 (1 = yes; 0 = no)	0.49	0.52	0.027 **
	(0.50)	(0.50)	(0.04)
(4) If the student has myopia (1 = yes; 0 = no)	0.16	0.12	−0.04 ***
	(0.37)	(0.33)	(0.00)
(5) Standardized mathematics scores at baseline	0.05	−0.14	−0.18 ***
	(0.98)	(1.02)	(0.00)
(6) Distance from the school to the student’s resident county (km)	34.31	34.11	−0.20
	(21.19)	(21.20)	(0.81)
Family characteristics			
(7) ln (family assets)	9.66	9.60	−0.05 **
	(0.96)	(0.96)	(0.03)
(8) Father’s education level (1 = is at least a high school graduate; 0 = lower than high school)	0.13	0.13	−0.00
	(0.34)	(0.34)	(0.73)
(9) Mother’s education level (1 = is at least a high school graduate; 0 = lower than high school)	0.79	0.80	0.01
	(0.41)	(0.40)	(0.14)
(10) Both father and mother migrate to urban areas for work (1 = yes; 0 = no)	0.12	0.13	0.01
	(0.33)	(0.34)	(0.23)
*N*	16,685	3909	

Note: *** *p* < 0.01, ** *p* < 0.05, * *p* < 0.1. SD, standard deviation. Data source: Authors’ survey.

**Table 2 ijerph-17-08200-t002:** Boarding status percentages.

Boarding Status	Boarding School Students in 2013		Total
	Non-Boarding	Boarding	
Non-boarding in 2012	12,618 (75.62%)	1020 (6.11%)	13,638 (81.74%)
Boarding in 2012	499 (2.99%)	2548 (15.27%)	3047 (18.26%)

Data source: Authors’ survey.

**Table 3 ijerph-17-08200-t003:** Comparison of the characteristics of students with different boarding status at baseline.

Control Variables	Total	Non-Boarding	From Non-Boarding to Boarding	H0: (2) = (3) Difference
	Mean	Mean	Mean	Mean
	(SD)	(SD)	(SD)	(*p*-Value)
	(1)	(2)	(3)	(4)
Students’ characteristics				
(1) Age (1 = at least 10 years old; 0 = less than 10 years old)	0.83	0.83	0.84	0.02
	(0.38)	(0.38)	(0.3666)	(0.35)
(2) Gender (1 = male; 0 = female)	0.51	0.50	0.55	0.043 ***
	(0.50)	(0.50)	(0.4981)	(0.01)
(3) If the student is in grade-4 (1 = yes; 0 = no)	0.50	0.50	0.49	−0.02
	(0.50)	(0.50)	(0.50)	(0.45)
(4) If the student has myopia (1 = yes; 0 = no)	0.15	0.15	0.16	0.01
	(0.36)	(0.36)	(0.37)	(0.47)
(5) Standardized mathematics scores at baseline	0.05	0.06	−0.13	−0.20 ***
	(0.97)	(0.97)	(1.00)	(<0.01)
(6) Distance from the school to the student’s resident county (km)	32.74	32.09	40.77	8.6808 ***
	(20.67)	(20.44)	(21.70)	(<0.01)
Family characteristics				
(7) ln (family assets)	9.60	9.58	9.8710	0.2938 ***
	(0.94)	(0.93)	(1.02)	(<0.01)
(8) Father’s education level (1 = is at least a high school graduate; 0 = lower than high school)	0.14	0.14	0.14	0.00
	(0.35)	(0.35)	(0.35)	(0.72)
(9) Mother’s education level (1 = is at least a high school graduate; 0 = lower than high school)	0.80	0.80	0.74	−0.06 ***
	(0.40)	(0.40)	(0.44)	(0.00)
(10) Both father and mother migrate to urban areas for work (1 = yes; 0 = no)	0.13	0.13	0.10	−0.03 ***
	(0.33)	(0.34)	(0.30)	(0.01)
*N*	16,685	12,618	1020	13,638

Note: *** *p* < 0.01, ** *p* < 0.05, * *p* < 0.1. SD, standard deviation. Data source: Authors’ survey.

**Table 4 ijerph-17-08200-t004:** The effect of boarding on students’ mental health.

Depedent variable: $\Delta y_{is}=y_{is,endline-}y_{is,baseline}$	ATT ^a^	Standard Errors (SEs)	*t*-Value
(1) Mental health	0.02	(0.05)	0.42
(2) Learning anxiety	−0.17	(0.15)	−1.11
(3) Anxiety about people	−0.10	(0.13)	−0.75
(4) Loneliness tendency	0.32 ***	(0.11)	3.01
(5) Self-blame tendency	0.05	(0.13)	0.40
(6) Allergy tendency	−0.04	(0.13)	−0.34
(7) Physical symptoms	0.10	(0.14)	0.70
(8) Horror tendency	0.07	(0.13)	0.52
(9) Impulsive tendency	0.02	(0.13)	0.16
*N*	16,685	16,685	16,685

Note: *** *p* < 0.01, ** *p* < 0.05, * *p* < 0.1. Standard errors (SEs) were calibrated by bootstrap (100 times). The table shows the regression results of boarding on mental health level and its different dimensions. Each regression equation controlled the personal characteristics of students (including gender, age, grade, myopia, standardized mathematics scores of students at baseline, and distance from school to county town) and family background (including natural logarithm of family finance, education level of parents, and whether parents migrate for work). ^a^ ATT, average treatment effect on treated, indicating the real effect of boarding on students’ mental health. Data source: Authors’ survey.

**Table 5 ijerph-17-08200-t005:** The effect of boarding on the mental health of students from different groups.

$Dependent\;variable:\Delta MHT_{is}=MHT_{is,endline}\;-\;MHT_{is,baseline}$	(1)	(2)	(3)
	ATT ^a^	SE	*t*-Value
Grade	−0.19 **	(0.09)	−2.13
(1) The effect of boarding for fourth-grade students	−0.13 *	(0.07)	−1.89
(2) The effect of boarding for fifth-grade students	0.13 **	(0.06)	2.15
If parents both migrate for work	0.31 **	(0.13)	2.4
(3) The effect of boarding for families with both parents migrating for work	0.30 **	(0.15)	1.97
(4) The effect of at least one parent not migrating for work and staying at home	−0.03	(0.05)	−0.75
If students are left-behind children (LBC) (5) The effect of boarding for LBC	0.29 **	(0.13)	2.16
*N*	16,685	16,685	16,685

Note: *** *p* < 0.01, ** *p* < 0.05, * *p* < 0.1 Standard errors were calibrated by bootstrap (100 times). The table shows the regression results of the heterogeneity analysis for different groups. Each regression equation controlled the personal characteristics of the students (including gender, age, grade, myopia, standardized mathematics scores of the students at baseline, and the distance from the school to the county) and family background (including the natural logarithm of family finance, the educational level of parents, and whether parents migrate for work). ^a^ ATT, average treatment effect for the treatment, indicating the real effect of boarding on students’ mental health. Data source: Authors’ survey.

**Table 6 ijerph-17-08200-t006:** Heterogeneity analysis of the effect of boarding on students’ mental health.

$Dependent\;variable:\;\Delta MHT_{is}=MHT_{is,endline}\;-\;MHT_{is,baseline}$	(1)	(2)	(3)
	ATT ^a^	SE	*t*-Value
(1) Differences between students aged 10 or more and students aged 10 or less	0.16	(0.11)	1.49
(2) Differences between male and female students	0.01	(0.08)	0.07
(3) Differences between grade 4 and grade 5 students	−0.20 **	(0.08)	−2.60
(4) Differences between myopic students and non-myopic students	0.11	(0.12)	0.91
(5) Differences between students whose distance from their school to their county town is greater than or equal to 32 km and students whose distance is less than 32 km ^b^	−0.12	(0.08)	−1.62
(6) Differences in student’s father educational level (whether above high school)	0.01	(0.12)	0.06
(7) Differences in student’s mother educational level (whether above high school)	−0.05	(0.11)	−0.43
(8) The differences between students with parents migrating for work and students with one parent not migrating for work	0.28 **	(0.14)	2.04
(9) The difference between the students whose family assets are in the top 50% of the broader population and the students in the bottom 50%	0.06	(0.06)	0.88
*N*	16,685	16,685	16,685

Note: *** *p* < 0.01, ** *p* < 0.05, * *p* < 0.1. Standard errors were calibrated by using the bootstrap method (100 times). The table shows the regression results of the heterogeneity analysis for different groups. Each regression equation controlled the personal characteristics of the students (including gender, age, grade, myopia, standardized mathematics scores of the students at baseline, and the distance from the school to the county) and family background (including the natural logarithm of family finance, the educational level of parents, and whether parents migrate for work). ^a^ ATT, average treatment effect for the treatment, indicating the real effect of boarding on students’ mental health. ^b^ Fifty percent of the students in the sample had a distance of more than 32 km from their school to the county, while the other 50% had a distance of less than 32 km from their school to the county. Data source: Authors’ survey.

**Table 7 ijerph-17-08200-t007:** The effect of boarding on students’ mental health from heterogeneity of grade 4 and grade 5.

Depedent Variable: $\Delta y_{is}=y_{is,endline}\;-\;y_{is,baseline}\;\;$	ATT ^a^	Standard Errors (SEs)	*t*-Value
(1) Mental health	−0.20 **	(0.08)	−2.60
(2) Learning anxiety	−0.39 **	(0.18)	−2.22
(3) Anxiety about people	−0.27 **	(0.13)	−2.06
(4) Loneliness tendency	0.05	(0.12)	0.39
(5) Self-blame tendency	−0.23	(0.14)	−1.64
(6) Allergy tendency	−0.18	(0.13)	−1.40
(7) Physical symptoms	−0.09	(0.16)	−0.59
(8) Horror tendency	0.03	(0.14)	0.18
(9) Impulsive tendency	−0.06	(0.13)	−0.44
*N*	1910	1910	1910

Note: *** *p* < 0.01, ** *p* < 0.05, * *p* < 0.1. The table shows the regression results of boarding on mental health level and its different dimensions with difference in grade 4 and grade 5. Each regression equation controlled the personal characteristics of students (including gender, age, grade, myopia, standardized mathematics scores of students at baseline, and distance from school to county town) and family background (including natural logarithm of family finance, education level of parents, and whether parents migrate for work). ^a^ ATT, average treatment effect on treated, indicating the real effect of boarding on students’ mental health. Data source: Authors’ survey.

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
