# Peer review of "The Effect of Boarding on the Mental Health of Primary School Students in Western Rural China"

_ijerph, 2020, doi:10.3390/ijerph17218200_

Round 1

Reviewer 1 Report

This manuscript needs to be more refined and corrected to improve before proceed it further. I hope that you have check the English editing, and attach verification of editing. For your guidance, I append the reviewers' comments below. 

  1. Most of all, the coherence and logicality are required in this manuscript. For this, please provide 'the purpose of this study' more clearly and specifically. 
  2. [Introduction section]: Please focus on and provide the main issues and reasons why this research is needed. I would like to invite the authors to provide more information about the differences between this study and previous studies. The introduction should start with a discussion of the scope and significance of the issue and or problem. Next, the manuscript needs a review of the literature that should provide the reader with a synthesis of previous work.
  3. [Introduction section]: Please describe why DID was used to identify the effect of boarding. Please describe specific reasons or evidence. 
  4. [Results]: The results section is difficult to follow. Logical reasoning is too weak in every phase of research methodology. There is need for more careful interpretation of the theoretical phase, fieldwork phase, and final analytical phase. Specially,
  5. [Discussion]: I would like to invite the authors to discuss more in-depth based on the results. There is need to clearly point out the main results, and interpret them in the light of earlier literature and discuss the results. There are some new articles to improve the discussion section. Please add them.

  6. References are outdated. Consider some latest studies.

Author Response

The Effect of Boarding on Mental Health of Primary Students in Western Rural China

Manuscript ID: ijerph-957308

Comments and Suggestions for Authors

This manuscript needs to be more refined and corrected to improve before proceed it further. I hope that you have check the English editing and attach verification of editing.

For your guidance, I append the reviewers' comments below.

Author’s response:

Thank you for reviewer’s remarks. We have gone through the paper carefully, referred to MDPI English editing and found native speakers to make good editing of the whole paper, including English editing, words ordering, sentence ordering, adding more logical transitions. Please see the editing tracks of new uploaded manuscript.

R1, Comment 1:

Most of all, the coherence and logicality are required in this manuscript. For this, please provide the purpose of this study' more clearly and specifically.

Response to R1, Comment 1:

                We apologize that we didn’t make the purpose of this study clear. We have added more sentences to make good coherence and logicality. We have made clear statements of study purpose.

                Revised parts:

Line 64-76

Although it is relatively easy to monitor the physical health status of students through more frequent and regular school physical examinations, it is more difficult to detect levels and problems of mental health [11]. A series of devastating suicides cases due to mental health problems from students in the early 2000s have prompted China’s government to make considerable efforts to reduce mental health problems in schools at all levels [12,13]. China’s government attaches great importance to the development of mental health among adolescents. The "Guidance on Strengthening Mental Health Services" report issued in December 2016 pointed out that "in order to comprehensively strengthen mental health education for children and adolescents, primary and secondary schools should pay attention to students' mental health education, cultivate positive, optimistic and healthy psychological qualities, and promote the sustainable development of students' mind and body” [14]. The rapid development of the boarding system in rural primary and secondary schools in China has increased concern and discussion regarding boarding school students' mental health.

Abstract and Line 145-149

Based on the panel data of 20,594 fourth- and fifth-grade students in the western provinces A and B, this paper used a combination of Propensity Score Matching (PSM) and the Difference-in-Differences (DID) model to correct the endogenous problems that may exist in the model, and to provide evidence for the effect of boarding on the mental health of rural primary school students.

R1, Comment 2:

  1. [Introduction section]: Please focus on and provide the main issues and reasons why this research is needed. I would like to invite the authors to provide more information about the differences between this study and previous studies. The introduction should start with a discussion of the scope and significance of the issue and or problem. Next, the manuscript needs a review of the literature that should provide the reader with a synthesis of previous work.

Response to R1, Comment 2:

                We fully agree with the comment and made revision from 3 aspects:

  1. The introduction starts with a discussion of the scope and significance of the issue and or problem. We are trying to point out the significance and background of boarding schools.

Revised parts:

Line 27-63

For individuals, receiving good education is conducive to the improvement of individual human capital and income level; For the country, the development of education is also conducive to the overall level of human capital and economic transformation and development.[1,2] Many developing countries in the world have attached importance to the construction of school institutions at the basic education stage. For example, Duflo (2001) has conducted an empirical study on the large-scale establishment of schools in Indonesia in the last century showing that if there is one more primary school for every 1,000 children on average, the average years of schooling will increase by 0.12–0.19 years [3]. Berlinski, Galiani and Gertler (2009) based on Argentina's study also found that the construction of pre-school education institutions significantly contributed to primary school students' academic performance [4].

China's rapid progress in education in recent decades benefited from China's long-term investment in basic education. In the 1990s, China implemented the national compulsory education project, and basically realized the pattern of "one village, one school" nationwide. However, since the mid-1990s, the number of school-age children in rural areas continued to decline, "one village, one school" school scale becoming small, high management costs, staffing difficulties and other issues were increasingly prominent [5]. The change of population situation brought about increase of management cost and difficulty of rural primary schools, which became the motivation of rural school layout adjustment. In 2001, China’s government issued the "Decision of the State Council on the Reform and Development of Basic Education" and proposed "adjusting the layout of rural compulsory education schools in line with local conditions", which started a new round of adjustment of the layout of primary and secondary schools in rural areas, also known as "withdrawal and consolidation of schools" [6].

In 2001, the State Council's Decision on Reform and Development of Basic Education pointed out that the layout of rural compulsory education schools should be adjusted according to local conditions [7]. Since then, the movement "withdrawal and consolidation of schools" began and swept across China’s rural primary and secondary schools. The implementation of this policy to some extent did integrate rural education resources, reduce the cost of education per student, but also increased the number of boarding schools [8]. Since 2001, with the continuous development and in-depth reform of China's education, as well as the needs of a large number of left-behind children in rural areas whose parents leave to work in urban areas, boarding schools have developed rapidly in rural areas of China and thus the number of boarding school students has quickly increased [9]. By the end of 2015, the total number of boarding school students across rural primary and secondary schools reached 26.363 million. The boarding rates of primary school and junior high school students were 14.4% and 58.6%, respectively. In the western rural areas, the proportion of primary school boarders reached 21.1% [10].

We fully agree with the comment and made revision from 3 aspects:

  1. This paper has provided more information about the differences between this study and previous studies.

Revised parts:

Line 126-144

Overall, the research literature in this area is quite rich, but there are still a few obvious problems that need to be further solved: First, the research methods used are mostly one-dimensional or multiple regression models and cross-sectional data are mostly used, which fails to solve the endogeneity problem caused by the self-selection bias and the omitting relevant variable bias. For example, Ma et al. (2013) and other researchers studied only 900 fifth- and sixth-grade students and adopted a simple variance analysis method. Second, the definition of boarding in most of the literature only examines whether students are boarding at a certain time, and does not consider the change in boarding status of students across a period of time. For example, Shen et al. (2012) used multiple regression and distinguished between boarding and non-boarding school students purely based on each student's boarding status during the survey. Since the time at which different rural children can board is different, it is reasonable to suspect that those students who board earlier are likely to experience other exogenous events in later school life. Without identification and divestiture, it is likely that the negative effects of boarding behavior will be overestimated [27]. Another important deficiency is that the existing research samples are mostly from either a single or a few schools in a single province, and the sample sizes are small and non-representative. This paper will try to cope with the deficiency list above using large, representative data and combination of Propensity Score Matching (PSM) and the Difference-in-Differences (DID) model to overcome endogenous problems.

We fully agree with the comment and made revision from 3 aspects:

  1. We have enriched the review of literature providing the readers with a synthesis of previous work. We added some updated literature. The previous version has only 22 references and now we have added to 43 references. Please see references parts.

Revised parts:

Line 77-125

Theoretically, boarding has both beneficial and detrimental effects on adolescent growth. On the positive side, boarding, a kind of collective life, to some extent, may reduce the risk of psychological problems on students. For example, group living environment supervised by school-assigned student caregivers may help maintain the health and safety of boarding students [15]. For those students who do not have enough family care, they may be able to get better care and life at school than in their families, which may contribute to their healthy growth [16]. Moreover, psychological counseling from teachers and communication with peers can help disadvantaged students to overcome the troubles and psychological problems and other common challenges experienced by students [17]. On the negative side, boarding implying that lack of care from family side may worsen students' mental health. Students in the primary school education stage are in a critical period of growth and development and boarding school students are separated from their parents for a long time, thus it is difficult for them to receive daily care from their parents and families during the period of time [18]. The collective living environment of boarding schools is also likely to resulting in students being more affected by the bad habits and behaviors of other students, and even makes them more susceptible to suffering from bullying , both of which would have a long-term negative impact on students' physical and mental development [19,20]. Serious mental health problems may hurt students' academic performance, which is not conducive to the improvement of China's human capital [21].

In recent years, the number of boarding schools in China has increased rapidly and a few studies have paid attention to the effect of boarding on students’ mental health status. However, from empirical literature, empirical research on the impact of boarding on students' mental health is inconsistent. A couple of studies have shown that the impact of boarding on school students’ mental health is negative. Evans-Campbell et al. (2012) believed that individuals that have been enrolled in boarding schools or have been raised by a person attending a boarding school are more likely to develop significant anxiety disorders and post-traumatic stress disorder than other individuals, and are even more likely to develop suicidal ideations [22]. Another study has shown that boarding hurts students' mental health, and that the negative effects do not disappear due to short-term [23]. Ma et al. (2013) found that the overall mental health of boarding school students was poor by analyzing the variances in fifth- and sixth-grade primary school students. Primary school students' self-blame tendency, learning adaptation independence, learning anxiety, allergy tendency, physical symptoms, and mental health test (MHT) total scores were significantly higher than those of non-boarding primary school students [24]. Chen et al. (2020) adopted instrumental variable to examine the effect of boarding and found to have negative effects on a few dimensions of students’ mental health (i.e. study anxiety, social anxiety, self-punishment, physical anxiety symptoms, and fear) reaching at 0.455 SDs (standard deviations) [11]. In contrast, some studies have shown that boarding either has no significant effect or has uncertain effects on the mental health of students. For example, Shen et al. (2012) used first- and second-year students in junior high school as research samples. He used multiple regression analysis to determine that there was no significant difference in the average scores of the full-scale MHT(Mental Health Test) and the eight subscales between boarding and non-boarding school students indicating that there was no significant difference in the mental health status between boarding and non-boarding school students [25]. Liu and Villa (2020) detected that though students who board in schools have improved their academic scores, their mental health level changed little [26]. Liu et al. (2004) used the psychotic symptom self-assessment form (SLE-90) to evaluate boarding and non-boarding high school students. They found that although boarding school students have more psychological problems in the initial enrollment stage, the psychological problems in the upper grades gradually decrease and the psychological health increases [27].

R1, Comment 3:

  1. [Introduction section]: Please describe why DID was used to identify the effect of boarding. Please describe specific reasons or evidence.

       Response to R1, Comment 3:

We value this suggestion and add more description about DID and PSM. Since introduction now has been quite long, we added more detailed in methods parts. Please see line 258-305.

                                Revised parts:

  1. Method

3.1. Propensity score matching (PSM)–Difference-in-Differences (DID) Method

According to Table 2, there were significant differences in individual characteristics and family economic background between the students in the treatment and control groups. These differences led to endogenous problems in the study, such as missing variable errors caused by unobservable factors and self-selection errors caused by observable factors. Therefore, this paper attempted to solve the above two problems by using a combination of propensity score matching (PSM) and the difference-in-differences (DID) model.

DID method is a quantitative statistical method, which is applied to at least two periods of data. It simulates the samples with changed research behavior between two periods of data as the "experimental group" in the real experiment and the samples without changed as the "control group" in the real experiment to explore the impact of the changes in research behavior on the outcome variables [29,30]. Specifically, in this study, when explore that effect of boarding on students' mental health, we chose students who did not board in the baseline period as the analysis sample (N=13,638), and divide this sample into two groups: one group was the students who changed from non-boarding to boarding, as the "experimental group" (N = 1020); the other group is students remaining non-boarding status, as the "control group" (N=12,618). The idea of applying DID method is to compare the difference in the change of an output variable between the experimental group and the control group before and after the implementation of a policy or before and after a certain behavior change.

The change in boarding behavior may be self-selection rather than random occurrence. If it is not treated and the sample is directly regressed, the parameter estimation will be biased. Economists proposed the propensity score matching method to reduce the error problem in the observation data set [31]. To avoid possible selection bias, propensity score matching has been used in this paper. The basic idea of the PSM is to find one or several control group students with similar or even the same endowment characteristics for each experimental group student through PSM, and then to compare the mean values of the result variables of the experimental and control groups under by controlling the other variables to obtain the estimation of the impact of boarding on the students' mental health [32]. The estimation steps involve: (1) the propensity scores of the students were estimated to establish the experimental group; (2) according to the common support of the propensity scores, the students in the experimental and control groups were matched by the closest matching method; (3) a balance test was carried out on the two matched groups of samples.

In this paper, the DID method controls the influence of unobservable variables through the difference in mental health before and after the change of boarding between the experimental group and the control group, especially the influence of factors that do not change over time and synchronously change over time, so as to effectively evaluate the net effect of boarding on students’ mental health. However, since the experimental group and the control group had different student and family characteristics, which does not meet the common trend premise assumption of using DID method, it is necessary to select the samples with similar characteristics in the control group and the experimental group as the counterfactual before performing DID. PSM can weaken selection bias and can obtain comparable samples of treatment and control groups. However, only observable factors can be controlled, and there are still missing variables for unobservable variables [33]. Therefore, this paper combined the PSM and DID methods to reduce endogeneity, since the DID method can control unobservable individual effects that are invariant over time. DID was used to estimate the average treatment effect of boarding on students' mental health based on the PSM results, with the differences in the standardized MHT scores as the explained variable. Therefore, this paper uses PSM-DID to estimate the average effect of boarding on students’ mental health.

R1, Comment 4:

  1. [Results]: The results section is difficult to follow. Logical reasoning is too weak in every phase of research methodology. There is need for more careful interpretation of the theoretical phase, fieldwork phase, and final analytical phase.

                Response to R1, Comment 4:

We highly value this suggestion and revised most parts of results.

First, we list clearly how the methods can be applied in the research.

Second, we elaborate the results with more relevant literature and explanation. We add 2 figures (Figure2 and Figure3) and 2 more tables (Table 6 and Table 7) to support our point of view and provide more convincing evidence.

Third, we revised some interpretation of results and put more explanation and backed data to support the main results.

                Revised parts:

Line 285-289

       The estimation steps involve: (1) the propensity scores of the students were estimated to establish the experimental group; (2) according to the common support of the propensity scores, the students in the experimental and control groups were matched by the closest matching method; (3) a balance test was carried out on the two matched groups of samples.

Line 343-382

After matching based on the personal and family characteristics of the students, the changes in the MHT standardized scores of the students in the experimental and control groups were found to differ by a standard deviation of 0.02 (Table 3, Row 1, Column 1), but the difference was not significant. This shows that boarding has no significant effect on students' mental health. According to the various dimensions, there was a significant difference between the students in the experimental group and the students in the control group only in the standardized scores of loneliness tendency, differing by a standard deviation of 0.32 (Table 3, Row 4, Column 1). This shows that after boarding, there is no significant change in the other mental health dimensions except that the loneliness tendency increased after boarding.

As for this result, there are three possible explanations.

First, in recent years (before or around 2012), China has attached great importance to the development of the mental health of primary and secondary school students and has increased the investment in boarding schools. For example, a few policies have been implemented to help students and schools to build up mental health education, such as “Guidelines for Mental Health Education in Primary and Secondary Schools” issued by Ministry of Education in 2002 and 2012. This policy helps teachers to help students with mental health disorders recover and adjust, especially for boarding students. Schools often play a substitute role for parents in terms of supervision and psychological counseling of students to some extent. However, due to students' natural psychological attachment to their parents, schools cannot completely replace the role of parents. Studies such as that of Li et al. (2015) have shown that parent–child attachment has a direct impact on boarding school students' adaptation to school [34]. Parent–child attachment refers to an intimate and lasting emotional connection with parents, which can provide support, security, and self-confidence for individuals. Boarding school students are far away from their parents and family for a long time and lack security, which makes it easy for them to feel lonely, which is a kind of explanation why loneliness tendency is significant higher.

Second, left-behind children accounted for a high proportion. In 2012, there were about 165 millions migrants and 44.76% of Chinese left-behind children (LBC) aged 12 to17 lived with their grandparents [35]. If we analyze the impact of boarding and left-behind children, as shown in Table 5 last row, the mental health of boarding LBC is worse than that of non-borders, which has an impact of 0.29 standard deviation and is significant at the confidence interval of 5%. The result implies if one parent goes out for working and students are boarding, their mental health is worse than who are not boarding, which also shows that if parents are not at home, students are prone to psychological problems such as loneliness. The above result is consistent with a latest study published in 2020. Chen et al. (2020) using data of 7,606 rural students prove that the effect of boarding is significantly higher among disadvantageous students. This study uses Instrumental Variable to examine the casual effect.

Third, in order to accurately identify the causal effect of boarding, this paper regards boarding as an intervention, which merely lasts one year. Short boarding period (one-year boarding) may not be enough for students' mental health change.

                Line 434-446

To explore the possible explanation, we made regression of the boarding effect on sub-dimensions of mental health between grade 4 and grade 5 as shown in Table 6. It can be seen that among students who have been boarding, in the dimension of learning anxiety and anxiety about people, the anxiety level of the fifth grade students is obviously higher than that of the fourth grade students, amounting to 0.39 SD and 0.27 SD. This may be because with the growth of grades, students find it more difficult in learning the courses and the fifth-grade students are facing greater pressure than the fourth-grade students. Studies have proved in rural schools, students do not have solid learning foundation. With the growth of grades, the learning content becomes more and more complex, the knowledge that students are not able to fully master is increasing, and students' learning anxiety is becoming more serious [38]. Additionally, fifth-grade students are facing great pressure to enter junior high school, and their mental health level is poorer [35]. Therefore, the mental health level of fifth-grade students deteriorates significantly after boarding.

R1, Comment 5:

  1. [Discussion]: I would like to invite the authors to discuss more in-depth based on the results. There is need to clearly point out the main results, and interpret them in the light of earlier literature and discuss the results. There are some new articles to improve the discussion section. Please add them.

Response to R1, Comment 5:

Thank you for gorgeous suggestions for improving the in-depth discussion. We have followed reviewer’s idea to interpret results with earlier literature and to make more clear and convincing discussion. Please see details in manuscript or below.

Revised parts:

Line 353-409

As for this result, there are three possible explanations.

First, in recent years (before or around 2012), China has attached great importance to the development of the mental health of primary and secondary school students and has increased the investment in boarding schools. For example, a few policies have been implemented to help students and schools to build up mental health education, such as “Guidelines for Mental Health Education in Primary and Secondary Schools” issued by Ministry of Education in 2002 and 2012. This policy helps teachers to help students with mental health disorders recover and adjust, especially for boarding students. Schools often play a substitute role for parents in terms of supervision and psychological counseling of students to some extent. However, due to students' natural psychological attachment to their parents, schools cannot completely replace the role of parents. Studies such as that of Li et al. (2015) have shown that parent–child attachment has a direct impact on boarding school students' adaptation to school [34]. Parent–child attachment refers to an intimate and lasting emotional connection with parents, which can provide support, security, and self-confidence for individuals. Boarding school students are far away from their parents and family for a long time and lack security, which makes it easy for them to feel lonely, which is a kind of explanation why loneliness tendency is significant higher.

Second, left-behind children accounted for a high proportion. In 2012, there were about 165 millions migrants and 44.76% of Chinese left-behind children (LBC) aged 12 to17 lived with their grandparents [35]. If we analyze the impact of boarding and left-behind children, as shown in Table 5 last row, the mental health of boarding LBC is worse than that of non-borders, which has an impact of 0.29 standard deviation and is significant at the confidence interval of 5%. The result implies if one parent goes out for working and students are boarding, their mental health is worse than who are not boarding, which also shows that if parents are not at home, students are prone to psychological problems such as loneliness. The above result is consistent with a latest study published in 2020. Chen et al. (2020) using data of 7,606 rural students prove that the effect of boarding is significantly higher among disadvantageous students. This study uses Instrumental Variable to examine the casual effect.

Third, in order to accurately identify the causal effect of boarding, this paper regards boarding as an intervention, which merely lasts one year. Short boarding period (one-year boarding) may not be enough for students' mental health change.

Though the above conclusions are contrary to the research results of the existing literature, there might be some deficiency of the previous research. Specially, most research on the relationship between boarding and students' mental health states that boarding has a negative effect on students' mental health, but such research has some problems, such as small sample sizes, unrepresentative samples, and unresolved endogeneity. For example, Zhang et al. (2009) used the mental health scale for middle school students as a test tool to compare the mental health status of 274 junior high school students in boarding schools and 300 junior high school students in non-boarding schools in Ningxia, China [36]. The study found that the overall psychological problems of junior high school students in boarding schools were significantly higher than those of non-boarding schools, and the detection rate of various psychological problems in the former was also higher than that of the latter. However, the sample size of this study was too small, and the method adopted was a simple t-test, which failed to solve the problems of endogeneity. Moreover, the research object of this paper was junior high school students, and there may be some differences between their mental health and that of primary school students. Another study from Wang and Mao (2015) analyzed the survey data of 8047 primary school students in grades 4, 5, 7, and 8 in 11 western regions [37]. They believed that boarding did not play a substitute role for family supervision for left-behind children, but rather became a negative factor affecting the development of left-behind children's social–emotional ability. Although the sample size was large and representative, the method used in this study was simple multiple regression analysis, and the mental health level of the students was tested through the self-compiled "Primary and Secondary School Students' Social Emotional Ability Questionnaire." Although the scale passed reliability and validity tests, the reliability of the scale needs further examination. This may be the deviation caused by different measuring tools. However, this paper has adopted a more convincing method PSM-DID with large and representative data, which can clearly estimate the real impact of boarding. We assume the results have been similar with most recent updated studies, such as Shi et al. (2016), Chen et al. (2020) and .Liu and Villa (2020) [11, 26, 35].

Line 432-464

First of all, according to the results in rows 1 and 2 of Table 5, the students in the experimental group of grade 4 had a better mental health level than those in the control group, while the students in the experimental group of grade 5 had a significantly poorer mental health level than those in the control group. That is, the mental health level of fourth-grade students improves after boarding, while the mental health level of fifth-grade students deteriorates. To explore the possible explanation, we made regression of the boarding effect on sub-dimensions of mental health between grade 4 and grade 5 as shown in Table 6. It can be seen that among students who have been boarding, in the dimension of learning anxiety and anxiety about people, the anxiety level of the fifth grade students is obviously higher than that of the fourth grade students, amounting to 0.39 SD and 0.27 SD. This may be because with the growth of grades, students find it more difficult in learning the courses and the fifth-grade students are facing greater pressure than the fourth-grade students. Studies have proved in rural schools, students do not have solid learning foundation. With the growth of grades, the learning content becomes more and more complex, the knowledge that students are not able to fully master is increasing, and students' learning anxiety is becoming more serious [38]. Additionally, fifth-grade students are facing great pressure to enter junior high school, and their mental health level is poorer [35]. Therefore, the mental health level of fifth-grade students deteriorates significantly after boarding.

Second, according to the results in rows 5 and 4 of Table 5, for families where at least one parent has not left to work in an urban area, there was no significant change in the mental health level of the students after boarding. However, for families whose parents are both migrant workers, the mental health level of students after boarding was significantly worse (SD = 0.30) at a significance level of 5%. If one parent is at home, boarders can avoid psychological problems by communicating with their parents when they return home on weekends. However, for families where both parents migrate to urban areas for work, students are not well cared for by their parents. After boarding, students are more likely to have psychological problems due to their reduced contact with their families, which is in line with literature in left-behind children, such as, Ge et al.(2015), Fellmeth et al.(2018), Jia and Tian(2010), Zhao and Yu(2016) [39-42]. There is also possibility that families where both parents need to go out have unobserved differences to other families (i.e., financial issues, opportunity structure). Overall, it might be that boarding can make up for the lack of family supervision of left-behind children to some extent, but it cannot completely replace the emotional communication of families.

R1, Comment 6:

  1. References are outdated. Consider some latest studies.

Response to R1, Comment 6:

As has been mentioned above, we have updated the references and referred to 43 references compared with the previous version 22 references. The latest references published after 2016 have been over 12 articles.

Revised parts:

Line 553-667

References:

  1. Heckman, James J., and Junjian Yi. Human capital, economic growth, and inequality in China. No. w18100. National Bureau of Economic Research, 2012.
  2. Autor, D. H., Levy, F., & Murnane, R. J. (2003). The skill content of recent technological change: An empirical exploration. The Quarterly journal of economics, 118(4), 1279-1333.
  3. Duflo, E. (2001). Schooling and labor market consequences of school construction in Indonesia: Evidence from an unusual policy experiment. American economic review, 91(4), 795-813.
  4. Berlinski, S., Galiani, S., & Gertler, P. (2009). The effect of pre-primary education on primary school performance. Journal of public Economics, 93(1-2), 219-234.
  5. Haepp, T., & Lyu, L. (2018). The impact of primary school investment reallocation on educational attainment in rural China. Journal of the Asia Pacific Economy, 23(4), 606-627.
  6. Liu, C., Zhang, L., Luo, R., Rozelle, S., & Loyalka, P. (2010). The effect of primary school mergers on academic performance of students in rural China. International Journal of Educational Development, 30(6), 570-585.
  7. State Council, 2001. Decisions on Reform and Development of Basic Education in China. Available at:(accessed on October 15, 2018) http://www.chinanews.com/2001-06-14/26/98195.html.
  8. Liu Xin.(2006). Layout adjustment of rural primary and secondary schools and construction of boarding schools. Education & Economy, (1):30-32.
  9. Guo Qingyang. On the Balanced Development of Compulsory Education and Construction of Rural Boarding Schools. Education & Economy, 2014 (4):36-43.
  10. Wu Zhihu and Qin Yuyou. (2017). Report on the Development of Rural Education in China, 2016. Beijing: Beijing Normal University Press.
  11. Chen, Q., Chen, Y., & Zhao, Q. (2020). Impacts of boarding on primary school students’ mental health outcomes–Instrumental-Variable evidence from rural northwestern China. Economics & Human Biology, 39, 100920.
  12. Hesketh, T., Ding, Q. J., & Jenkins, R. (2002). Suicide ideation in Chinese adolescents. Social psychiatry and psychiatric epidemiology, 37(5), 230-235.
  13. Zhang, L., Kleiman-Weiner, M., Luo, R., Shi, Y., Martorell, R., Medina, A., & Rozelle, S. (2013). Multiple micronutrient supplementation reduces anemia and anxiety in rural China's elementary school children. The Journal of nutrition, 143(5), 640-647.
  14. National Health of Commissions of the People’s Republic of China (NHC). (2017). Guidance on Strengthening Mental Health Services. Retrieved on http://www.nhc.gov.cn/jkj/s5888/201701/6a5193c6a8c544e59735389f31c971d5.shtml
  15. Wang, L., 2017. Exploring how to improve the quality of small rural schools—a summary of experiences from a bottom-up approach. In: Yang, D., Yang, M., Huang, S. (Eds.), Annual Report on China’s Education. Social Sciences Academic Press, Beijing, China (In Chinese). ISBN 978-7-5201-0552-1
  16. Liu, Y. M. (2005). Care for rural left-behind children. Journal of China Agricultural University, 60(3), 29-33.
  17. Granot, D., & Mayseless, O. (2001). Attachment security and adjustment to school in middle childhood. International Journal of Behavioral Development, 25(6), 530-541.
  18. Martin, A. J., Papworth, B., Ginns, P., & Liem, G. A. D. (2014). Boarding school, academic motivation and engagement, and psychological well-being: A large-scale investigation. American Educational Research Journal, 51(5), 1007-1049.
  19. Ak, L., & Sayil, M. (2006). Three Different Types of Elementary School Students' School Achievements, Perceived Social Support, School Attitudes and Behavior-Adjustment Problems. Educational Sciences: Theory & Practice, 6(2).
  20. Colmant, S., Schultz, L., Robbins, R., Ciali, P., Dorton, J., & Rivera-Colmant, Y. (2004). Constructing meaning to the Indian boarding school experience. Journal of American Indian Education, 22-40.
  21. Wang, H., Chu, J., Loyalka, P., Xin, T., Shi, Y., Qu, Q., & Yang, C. (2016). Can social–emotional learning reduce school dropout in developing countries?. Journal of Policy Analysis and Management, 35(4), 818-847.
  22. Evans-Campbell, T., Walters, K. L., Pearson, C. R., & Campbell, C. D. (2012). Indian boarding school experience, substance use, and mental health among urban two-spirit American Indian/Alaska Natives. The American Journal of Drug and Alcohol Abuse, 38(5), 421-427.
  23. Kleinfeld, J., & Bloom, J. (1977). Boarding schools: effects on the mental health of Eskimo adolescents. The American journal of psychiatry.
  24. Ma Xinyi, Ling Hui, Li Xinli and Wang Mengyi. (2013). Comparison of Learning Adaptability, Mental Health and Academic Achievement between Boarding and Non-Boarding Primary School Students. Chinese Journal of Clinical Psychology, 21(3): 497-499.
  25. Shen Zili, Chen Li, Cui Jianhua, Zhang Chuijing, Xie Xingzhi, Ji Xiaoxia, and Li Yan. (2012). Influencing factors of mental health of boarding and non-boarding junior high school students. Educational Measurement and Evaluation, (8): 38-42.
  26. Liu, M. , & Villa, K. M. . (2020). Solution or isolation: is boarding school a good solution for left-behind children in rural china?. China Economic Review
  27. Liu Chaojun, Tian Suying, Xun Guanglei, and Jia Zhanling. (2004). Comparison of mental health status of high school students in boarding and non-boarding schools. Chinese Journal of Tissue Engineering Research, 8(27): 5782-5784.
  28. Zhou Bucheng. (1991). Mental Health Diagnostic Test (MHT) Manual. Shanghai: Department of Psychology, East China Normal University.
  29. Lechner, M. (2011). The estimation of causal effects by difference-in-difference methods. Now.
  30. Wing, C., Simon, K., & Bello-Gomez, R. A. (2018). Designing difference in difference studies: best practices for public health policy research. Annual review of public health, 39.
  31. Caliendo, M., & Kopeinig, S. (2008). Some practical guidance for the implementation of propensity score matching. Journal of economic surveys, 22(1), 31-72.
  32. Heinrich, C., Maffioli, A., & Vazquez, G. (2010). A primer for applying propensity-score matching. Inter-American Development Bank.
  33. Dehejia, R. (2005). Practical propensity score matching: a reply to Smith and Todd. Journal of econometrics, 125(1-2), 355-364.
  34. Li Mian, Zhang Pingping, Zhang Xinghui, Wang Geng. (2015). Relationship between Parent-child Attachment and School Adaptation of Junior Middle School Boarding Students: Mediating Role of Separation and Individualization, Chinese Journal of Clinical Psychology, 23(04):718-721.
  35. Shi, Y., Bai, Y., Shen, Y., Kenny, K., & Rozelle, S. (2016). Effects of parental migration on mental health of left‐behind children: Evidence from northwestern China. China & World Economy, 24(3), 105-122.
  36. Zhang Lijin, Shen Jie, Li Zhiqiang, Gai Xiaosong. (2009). Comparison of mental health status of junior high school students in boarding and non-boarding schools. Chinese Journal of Special Education, 5:82-86.
  37. Wang Shutao, Mao Yaqing.(2015). The Influence of Boarding on the Development of Social Emotional Ability of Left-behind Children: An Empirical Study Based on 11 Western Provinces . Journal of Educational Studies, 2015(5):111-120.
  38. Wang, H., Yang, C., He, F., Shi, Y., Qu, Q., Rozelle, S., & Chu, J. (2015). Mental health and dropout behavior: A cross-sectional study of junior high students in northwest rural China. International Journal of Educational Development, 41, 1-12.
  39. Ge, Y., Se, J., & Zhang, J. (2015). Research on relationship among internet-addiction, personality traits and mental health of urban left-behind children. Global journal of health science, 7(4), 60.
  40. Fellmeth, G., Rose-Clarke, K., Zhao, C., Busert, L. K., Zheng, Y., Massazza, A., ... & Orcutt, M. (2018). Health impacts of parental migration on left-behind children and adolescents: a systematic review and meta-analysis. The Lancet, 392(10164), 2567-2582.
  41. Jia, Z., & Tian, W. (2010). Loneliness of left‐behind children: a cross‐sectional survey in a sample of rural China. Child: care, health and development, 36(6), 812-817.
  42. Zhao, F., & Yu, G. (2016). Parental migration and rural left-behind children’s mental health in China: A meta-analysis based on mental health test. Journal of Child and Family Studies, 25(12), 3462-3472.
  43. China Stata Concil.(2016). Opinions of the State Council on Strengthening the Care and Protection of Left-behind Children in Rural Areas. Retrieved on: http://www.gov.cn/zhengce/content/2016-02/14/content_5041066.htm

We are looking forward that this revised version can satisfy reviewer and the journal.

Thank you for your great efforts and contribution on the paper!

Reviewer 2 Report

The large and high-quality sample is praiseworthy.

More care needs to be exercised in the wording in the introduction. Sometimes it sounds as if boarding makes students so and so (i.e., causality) when the data are merely correlational.

In Section 2.1 much more details on sample mortality should be provided, for example, how many were lost at which state. Did the lost ones differ from the retained ones? At present, it only says: “part of the sample was lost”.

Section 2.2.1 How many students were excluded due to lying (that is, a score higher than 7 on the lying scale)?

There seems to be a critical typo: “The higher the total score, the worse the students' mental health.” In Figure 1 the lower the MHT score the lower mental health.

Section 2.2.2 Despite being a matter of definition, the baseline non-boarders should not be termed “experimental group”, as this is not an experiment (i.e., with random assignment). “Treatment group” would be less misleading.

Please explain: “To ensure that all students in the baseline have the same boarding situation, this paper removed 3,047 boarding students from the baseline.” Why was this needed? Who was removed by which method? Did the removed ones differ from the retained ones? Can anything meaningful be contributed by looking at substantive results from these excluded pupils? After all, that is a big group.

It did not become clear to me how exactly the two methods were combined (i.e., PSM and DID).

Section 4.1: The interpretation “The possible explanation for this result is that in recent years, China has attached great importance to the development of the mental health of primary and secondary school students and has increased the investment in boarding schools.” This is not backed by the data. You need a pre-post study to substantiate that claim.

The interpretation: “The possible explanation for this result is that teacher may think that the fourth-grade students are younger and they care more about the fourth-grade students” does not strike me as particularly plausible and there are no grounds given (for example, references or supplementary data from teacher interviews).

“However, for families where both parents go out, students are not well cared for by their parents. After boarding, students are more likely to have psychological problems due to their reduced contact with their families.” This interpretation is not backed by the data. It might be that families where both parents need to go out have unobserved differences to other families (i.e., financial issues, opportunity structure).

In general, many interpretations should be couched much more cautiously and/or backed up by references and additional information. Moreover, the patriotic tone in favor of China struck me as unusual in a slightly negative sense and should be toned down. A native-language language editor needs to go over the manuscript. In general, the manuscript lacks transparency and detail at several important places, but the substance (i.e., data and analyses) is good.

Round 2

Reviewer 1 Report

The explanations provided by the author are exhaustive. Furthermore, several substantial changes made to the manuscript have increased its quality. 

Reviewer 2 Report

I believe the manuscript has been significantly improved and now warrants publication in IJERPH (after correcting some remaining typos und presenting explanations more as suggestions and not as final and complete).